# Buserelin Acetate Added to Boar Semen Enhances Litter Size in Gilts in Tropical Environments

**DOI:** 10.3390/ani14172501

**Published:** 2024-08-28

**Authors:** Preechaphon Taechamaeteekul, Chatchapong Jaijarim, Chairach Audban, Kridtasak Sang-Gassanee, Pongsak Numsri, Hongyao Lin, Miquel Collell, Padet Tummaruk

**Affiliations:** 1Centre of Excellence in Swine Reproduction, Department of Obstetrics, Gynaecology and Reproduction, Faculty of Veterinary Science, Chulalongkorn University, Bangkok 10330, Thailand; taechamaeteekul.p@gmail.com; 2Charoen Pokphand Foods Public Company Limited, Bangkok 10120, Thailand; chatchapong.jai@cpf.co.th (C.J.); chairach.a@cpf.co.th (C.A.); 3Intervet (Thailand) Ltd., South Sathorn Rd., Yannawa, Sathorn, Bangkok 10120, Thailand; kridtasak.sang-gassanee@merck.com (K.S.-G.); pongsak.numsri@merck.com (P.N.); 4Merck Animal Health, 2 Giralda Farms, Madison, NJ 07940, USA; hongyao.lin@msd.com (H.L.); miquel.collell@merck.com (M.C.)

**Keywords:** artificial insemination, buserelin, litter size, pig, semen

## Abstract

**Simple Summary:**

The timing of ovulation varies widely among gilts, leading farms to often perform multiple inseminations to ensure that sperm and eggs meet in the reproductive tract, thus enhancing fertility. Administering a gonadotropin-releasing hormone (GnRH) agonist, such as buserelin, at the onset of estrus can induce ovulation and reduce the variation in ovulation timing among sows. This study examined the effect of adding buserelin to boar semen on the litter size of gilts. Gilts were inseminated with semen enriched with either 5 or 10 µg of buserelin acetate during the initial insemination. Although there were no significant differences in the farrowing rate between the groups, the group receiving 10 µg of buserelin acetate had a higher total number of piglets and live-born piglets. Additionally, this group exhibited greater litter birth weight compared to the control group. These findings indicate that supplementing boar semen with a GnRH agonist enhances litter size in gilts.

**Abstract:**

The use of exogenous hormones has long been of interest for improving reproductive performance in swine production. Enhancing litter size directly impacts the economic efficiency of pig production. Various strategies, including nutritional, genetic, and hormonal approaches, have been explored with varying degrees of success. Administering a gonadotropin-releasing hormone (GnRH) agonist, such as buserelin, at the onset of estrus can induce ovulation and reduce the variation in ovulation timing among sows. This study assessed the impact of GnRH agonist supplementation in boar semen doses on the litter size of inseminated gilts. The research was conducted on a commercial swine herd in northern Thailand. A total of 231 Landrace × Yorkshire crossbred gilts, aged 224.5 ± 16.2 days at the onset of estrus synchronization, participated in the experiment. The gilts’ estrus was synchronized with oral altrenogest supplementation at a dosage of 20 mg/day for 18 days. After exhibiting standing estrus, the gilts were randomly divided into three groups. Control group: gilts were inseminated at 0 and 12 h post standing estrus onset with a conventional semen dose (*n* = 94). Treatment 1: similar to the control group, but with an added 5 µg (1.25 mL) of buserelin acetate to the boar semen dose during the first insemination (*n* = 71). Treatment 2: similar to the control group, but with 10 µg (2.5 mL) of buserelin acetate added to the boar semen dose during the first insemination (*n* = 66). All gilts were inseminated twice during their standing estrus using the intrauterine artificial insemination method. Each semen dose contained 3.0 × 10^9^ motile sperm in 80 mL. The farrowing rate averaged 78.8% and did not significantly differ between the groups (*p* = 0.141). The total number of piglets born per litter in the treatment 2 group was greater than in the control group (14.0 ± 0.3 vs. 13.2 ± 0.3, respectively, *p* = 0.049), but was not significantly different from the treatment 1 group (13.3 ± 0.3, *p* = 0.154). Similarly, the number of live-born piglets in the treatment 2 group was greater than in the control and treatment 1 groups (13.2 ± 0.4 vs. 12.3 ± 0.3 and 12.0 ± 0.4, respectively, *p* < 0.05). Moreover, the live-born piglets’ litter birth weight in the treatment 2 group was greater than in the control group (17.0 ± 0.4 vs. 15.6 ± 0.3 kg, respectively, *p* = 0.008) and the treatment 1 group (15.7 ± 0.4 kg, *p* = 0.025). In conclusion, adding a GnRH agonist to boar semen appears to enhance the litter size of gilts. Further research should focus on understanding the underlying mechanisms and determining the optimal dose and timing for GnRH agonist supplementation.

## 1. Introduction

Artificial insemination (AI) is a common technique in the swine industry [1]. Due to various factors such as the sow’s nutritional status at weaning, the number of piglets weaned, the sow’s parity, backfat thickness, and the interval from weaning to estrus, ovulation timing can vary significantly among sows [2]. Consequently, farms often perform multiple inseminations to ensure the simultaneous presence of viable sperm and eggs in the reproductive tract [3]. Indeed, the ovulation process in sows occurs within a short period of time (i.e., within 3 h) [4], making it crucial to inseminate with a single dose of semen close to the time of ovulation [5]. Consequently, multiple inseminations (2–3 times during standing estrus) are not entirely necessary if the timing of ovulation can be controlled. Physiologically, aligning insemination with ovulation timing can enhance sow fertility [5]. For conception to occur, sperm must be present in the reproductive tract before ovulation, given the oocyte’s brief viability period [6]. Ideally, inducing ovulation at insemination could optimize reproduction in swine. Previous research has demonstrated that adding substances like estrogen [7], PGF2α [8], and oxytocin [9] to semen can increase farrowing rates and litter sizes by potentially enhancing uterine contractions, which facilitate sperm movement from the vagina to the uterus. In practice, mating can trigger the release of oxytocin from the sow’s central nervous system, increasing uterine activity due to the boar’s presence during AI [10]. Likewise, estrogen in boar seminal plasma can increase myometrial contractions by triggering the local release of PGF2α from the endometrium [11]. However, these outcomes are not always consistent, with more positive results typically observed during summer. Therefore, an alternative method to synchronize ovulation with insemination is needed.

Fixed-time artificial insemination (FTAI) programs, which employ pharmacological agents to induce ovulation for timed sperm introduction, are well-established in livestock management, addressing the challenge of unpredictable ovulation timing [12,13,14,15]. These protocols have seen considerable success and widespread adoption in the cattle industry [16] and hold potential for enhancing swine reproductive efficiency as well. Central to FTAI strategies is the utilization of gonadotropin-releasing hormone (GnRH) agonists, which are favored for their ability to stimulate a nearly natural luteinizing hormone (LH) surge. These agonists can be delivered through various methods, and the market offers a range of synthetic options with distinct bioactivities and durations of action [17]. Notable examples include buserelin [13], lecirelin [18], and triptorelin [19]. In the case of weaned sows, buserelin administration has been shown to prompt ovulation approximately 37.5 ± 3.3 h post-administration, which is significantly quicker than the average 63.6 ± 9.6 h it takes for natural ovulation to occur following the onset of standing estrus [20].

In practical settings on many farms, the use of GnRH agonists through traditional methods results in a wide range of ovulation timing success among sows, with rates varying from 57.9% [13] to as high as 89% [21]. The effectiveness tends to decrease for sows with follicles smaller than 6.5 mm at the time of GnRH agonist administration [22]. The application of these compounds in gilts presents additional challenges due to the absence of a natural event like weaning that would otherwise synchronize estrus, ensuring uniform follicle size across individuals. To address this, estrus in gilts often requires synchronization through artificial means, such as the administration of altrenogest [15]. Factors such as the timing and method of administration, as well as the number of previous parities, have been identified as contributing to the varied responses to GnRH agonists [23]. Consequently, there is a clear need to refine the use of GnRH agonists for improved consistency and effectiveness.

In rabbits, a species that requires induction for ovulation, the inclusion of GnRH agonists like buserelin and lecirelin in semen has been shown to enhance ovulation rates compared to controls [24]. To date, the study by Manjarin et al. [25] appears to be the only investigation into the impact of adding buserelin to semen on farrowing rates and litter sizes at birth within a commercial pig farming context. Their research demonstrated that adding 2 µg of buserelin (0.5 mL Receptal^®^, MSD Animal Health, Rahway, NJ, USA) to boar semen during the initial insemination boosted the farrowing rate from 84.9% in the control group to 93.2% in the treated group. Furthermore, the number of piglets born per litter increased from a range of 12.4–13.4 in the control group to 14.2–15.5 in the treated group [25].

To validate the effectiveness of interventions like buserelin supplementation in semen, large-scale clinical trials should be conducted under various farm management practices and climatic conditions to confirm result consistency. In tropical regions, swine tend to exhibit lower reproductive performance than those in temperate areas, likely due to the stress of higher temperatures on their endocrine systems [26]. Furthermore, gilts typically show more variability in ovulation timing and generally have lower reproductive success compared to more mature sows [27]. Therefore, examining the use of buserelin in semen within a tropical setting and specifically in gilts could represent a challenging “worst-case” scenario that has not yet been explored. Consequently, this study aimed to assess the impact of adding a GnRH agonist (buserelin) to boar semen during the initial AI on the conception rate, farrowing rate, and litter size at birth for gilts in hot temperature zones.

## 2. Materials and Methods

### 2.1. Animals

This study followed the ethical principles and guidelines for the use of animals for scientific purposes set by the National Research Council of Thailand, which was approved by the Institutional Animal Care and Use Committee in compliance with Chulalongkorn University’s regulations and policies on the care and use of experimental animals. (Approval number 2331044). The research was conducted on a commercial swine herd in northern Thailand from October 2022 to January 2023. A total of 231 Landrace × Yorkshire crossbred gilts, with an average age of 224.5 ± 16.2 days at the onset of estrus synchronization, participated in the experiment. The gilts were fed 2.5–3.0 kg/day of a corn and soybean-based diet containing 15.0% crude protein, 0.85% digestible lysine, and 3.0 Mcal/kg of metabolizable energy. They were housed in a closed system with an evaporative cooling system. In the closed housing system, the temperature inside the barn was maintained below 27 °C, while humidity was not regulated. During the experimental period, the temperatures inside the barn averaged 22.9 ± 3.6 °C, ranging from 14.5 to 32.0 °C. The average humidity levels were 88.8 ± 7.0%, with a range of 45.5 to 98.0%. At 8 months old, the gilts were moved from the gilt pool to the breeding house and placed in individual crates measuring 0.6 × 2.0 m on a concrete slatted floor. A prerequisite for this transition was the demonstration of estrus in the presence of a mature boar. For the breeding process, the gilts’ estrus cycles were synchronized through oral altrenogest (Regumate^®^, MSD Animal Health, NJ, USA) supplementation at a dose of 20 mg/day for 18 days. Mating was conducted once the gilts exhibited estrus symptoms following the conclusion of altrenogest treatment, provided they weighed at least 130 kg.

After insemination, all gilts were moved to the gestation house and kept individually in crates until one week before their expected farrowing date. Throughout gestation, their diet comprised 1.5–3.5 kg daily of a corn–soybean feed with 16.0% crude protein, 1.0% digestible lysine, and 2.8 Mcal/kg of metabolizable energy. Their feed intake gradually increased to 3.0–3.5 kg/day from weeks 12 to 15 of gestation, then slowly decreased in the final week before farrowing. About a week before their due date, the gilts were relocated to the farrowing house and placed in individual crates measuring 0.6 × 2.2 m within the larger farrowing pens, which measured 2.0 × 2.2 m. The pens were designed with a concrete base in the center for the sows and steel slats on the sides of the farrowing crate for the piglets. During lactation, the sows received 5.0–6.0 kg/day of a corn–soybean diet with 18.0% crude protein, 1.0% digestible lysine, and 3.2 Mcal/kg of metabolizable energy, provided in 3 meals per day. Water was always available through nipple drinkers. Backfat thickness was assessed using A-mode ultrasonography (Renco Lean-Meater^®^, Minneapolis, MN, USA) during the first insemination. Measurements were taken at the last rib, about 6–8 cm off-center on both sides of the sows, and the average of the left and right measurements was determined.

### 2.2. Estrus Detection, Artificial Insemination, Experimental Design, Pregnancy Detection, and Semen Evaluation

Twice daily, at 7 am and 4 pm, starting the day after their arrival in the gilt pool, trained personnel performed estrus detection on the gilts using the back pressure test in the presence of a boar stud. Gilts that exhibited a clear standing response when near the boar—indicated by accepting the back pressure test, having erect ears, and showing immobile legs—were recognized as being in estrus. For artificial insemination (AI), an intrauterine AI catheter was used to insert a semen dose of 3.0 × 10^9^ motile sperm in an 80 mL volume. The semen used for all the gilts was pooled from two to three Duroc boars, aged 1 to 3 years, and collected using the gloved hand technique. This semen was collected and processed daily at a boar stud, with each boar providing a semen sample every 5 to 7 days. For AI, pooled semen from two to three boars was used for each gilt. The total sperm motility of boar semen in the present study averaged 82.3 ± 4.3% (mean ± SD). Sperm concentration was assessed using a spectrophotometer (Spermacue^®^, Minitübe GmbH, Tiefenbach, Germany), and the ejaculates were diluted with a semen extender (MIII^®^, Minitübe International AG, Tiefenbach, Germany) to reach a final concentration of 37.5 × 10^6^ motile sperm per ml. The diluted semen was stored in a refrigerator at 17 °C and used within one day of collection, in accordance with herd regulations [15].

Animals were randomly divided into three groups: Control: gilts were inseminated at 0 and 12 h post standing estrus onset with a conventional semen dose (*n* = 94). Treatment 1: similar to the control group, but with an added 5 µg (1.25 mL, Porceptal^®^, MSD Animal Health, NJ, USA) of buserelin acetate to the boar semen dose during the first insemination (*n* = 71). Treatment 2: similar to the control but with 10 µg (2.5 mL, Porceptal^®^, MSD Animal Health) of buserelin acetate added to the boar semen dose during the first insemination (*n* = 66). All gilts were inseminated twice during their standing estrus using the intrauterine artificial insemination method. Each semen dose contained 3 billion (3.0 × 10^9^) motile sperm in 80 mL. For all groups, the second insemination, conducted 12 h post standing estrus, was performed with only a conventional semen dose. Conception rates were determined 28.0 ± 2.0 days after insemination with trans-abdominal ultrasonography (HS-2000, Honda Electronics Co., Ltd., Toyohashi, Aichi, Japan).

To assess the effect of buserelin on boar semen quality, an additional trial was conducted using 10 semen ejaculates from 10 Duroc boars. Each ejaculate was diluted with Beltsville Thawing Solution (BTS) and divided into three parts. These parts were then supplemented with either 0 (control), 5, or 10 µg of buserelin. Total sperm motility and progressive motility were measured within 20 min of adding buserelin, utilizing a computer-assisted sperm analysis system (SCA^®^ CASA System, MICROPTIC S.L., Barcelona, Spain).

### 2.3. Data

Reproductive performance metrics were analyzed, including the rate of regular return to estrus, conception rate at 28.0 ± 2.0 days post-insemination, and farrowing rate. The proportion of sows exhibiting a regular return to estrus was determined through estrus detection, facilitated by boar contact twice daily between 18 and 24 days post-insemination. The regular return rate was calculated as the number of gilts returning regularly to estrus divided by the total number of inseminated gilts, then multiplied by 100. Similarly, the farrowing rate was defined as the number of gilts that farrowed divided by the total number of inseminated gilts, multiplied by 100. In addition, litter characteristics at birth, including total piglets born per litter, number of live births per litter, proportions of mummified and stillborn piglets per litter, and overall litter birth weight, were assessed in 181 gilts (80, 51, and 50 gilts in control, treatment 1, and treatment 2, respectively). Additionally, piglet-specific characteristics—including individual birth weight, the coefficient of variation in birth weight within litters, and the percentage of piglets weighing less than 1000 g at birth—were randomly assessed across these litters (*n* = 1420 piglets). Piglet body weight was measured immediately post-birth using a digital scale (SDS^®^ IDS701-CSERIES, SDS Digital Scale Co., Ltd., Yangzhou, China). The total litter birth weight was determined by summing the individual weights of all piglets in a litter.

### 2.4. Statistical Analysis

Statistical analyses were performed using SAS version 9.4 (SAS Institute Inc., Cary, NC, USA). Descriptive statistics for the reproductive data of gilts in the control, treatment 1, and treatment 2 groups were calculated using the MEANS procedure. Frequency analyses for categorical traits, such as regular return-to-estrus rate, conception rate, and farrowing rate, were conducted using the FREQ procedure. These categorical data were further examined using logistic regression with the GLIMMIX procedure, incorporating factors such as group (control, treatment 1, treatment 2) into the model. Least-squares means for each category were calculated and compared using the least significant difference test to assess significance.

Continuous variables, such as the total and live number of piglets born per litter, percentages of mummified fetuses and stillborn piglets, gestation length, gilt age, backfat thickness at insemination, litter birth weight, and variation in piglet birth weights within litters, were analyzed using the general linear model (GLM) procedure. The analysis considered factors such as group (control, treatment 1, treatment 2), with least squares means calculated for each variable category and compared using the least significant difference test.

For analyzing piglet birth weight, a general linear mixed model (MIXED) procedure was employed, incorporating group (control, treatment 1, treatment 2) as a fixed effect and gilt identities number as a random effect. The proportion of piglets weighing less than 1000 g was presented as a percentage and analyzed using logistic regression via the GLIMMIX procedure. The total motility and progressive motility of boar semen supplemented with 0, 5, and 10 µg of buserelin were analyzed using the GLM procedure. Least-square means were calculated and compared using the least significant difference test with Tukey–Kramer adjustment for multiple comparisons. A *p*-value of less than 0.05 indicated statistical significance in all analyses.

## 3. Results

Table 1 presents the reproductive outcomes, including fertility and litter characteristics, of gilts following the addition of buserelin to boar semen at doses of 5 µg (treatment 1) or 10 µg (treatment 2), in comparison to traditional AI (control). The farrowing rate averaged 78.8% and did not significantly differ between the groups (*p* = 0.141). The farrowing rates in the control, treatment 1, and treatment 2 groups were 85.1%, 73.2%, and 75.8%, respectively (Table 1). In the treatment 2 group, the average total number of piglets born per litter was higher than in the control group (14.0 ± 0.3 vs. 13.2 ± 0.3, respectively, *p* = 0.049) but did not significantly differ from the treatment 1 group (13.3 ± 0.3, *p* = 0.154). The number of piglets born alive in the treatment 2 group also exceeded those in the control and treatment 1 groups (13.2 ± 0.4 vs. 12.3 ± 0.3 and 12.0 ± 0.4, respectively, *p* < 0.05). Furthermore, the litter birth weight of live-born piglets in the treatment 2 group was higher than that in the control group (17.0 ± 0.4 kg vs. 15.6 ± 0.3 kg, respectively, *p* = 0.008) and the treatment 1 group (15.7 ± 0.4 kg, *p* = 0.025). The litter birth weight of live-born piglets in the treatment 1 group was not significantly different compared to the control group, *p* = 0.862). Piglet characteristics were not significantly different across the groups (Table 1). However, the percentage of piglets weighing less than 1000 g in the treatment 2 group (11.1%) showed a trend towards being lower compared to the control (15.0%, *p* = 0.111) and treatment 1 (15.2%, *p* = 0.063) groups (Table 1).

In the analysis of boar semen quality, total sperm motility within 20 min after treatment with 0, 5, and 10 µg of buserelin did not differ significantly among the groups. The total sperm motility rates were 84.0 ± 1.3%, 82.3 ± 1.3%, and 80.5 ± 1.3%, respectively (*p* = 0.191). However, the progressive sperm motility of semen supplemented with 10 µg of buserelin significantly declined compared to the control group (37.7 ± 3.2% vs. 49.5 ± 3.2%, respectively, *p* = 0.033). Meanwhile, the progressive sperm motility of semen supplemented with 5 µg of buserelin did not differ significantly from the control group (42.6 ± 3.2%, *p* = 0.279).

## 4. Discussion

The primary objective of this study was to investigate the effectiveness of buserelin in inducing ovulation when combined with semen. Litter size at birth was the main indicator used for this purpose. Additionally, we assessed the number of live-born piglets and their birth weights. However, a limitation of our study is that we could not track piglet mortality rates during the lactation, nursery, and fattening periods due to management issues such as cross-fostering and mixing animals during the nursery phases. Therefore, the efficacy of buserelin in inducing ovulation in gilts can only be demonstrated by an increase in litter size at birth, which is also an important economic trait in pigs.

The secondary objective of the present study was to demonstrate the feasibility of diluting buserelin in semen and administering it into the uterus simultaneously with artificial insemination as an alternative to the traditional intramuscular injection route. This method was explored to reduce the need for injections and improve animal welfare in the swine industry. Intramuscular injections often cause stress and discomfort to animals, which can negatively affect their reproductive outcomes. Such injections should be limited to sick animals and not used routinely, such as for artificial insemination. Previous research has shown that another GnRH agonist, triptorelin acetate, can be administered intravaginally to induce ovulation in sows [12,22]. However, the feasibility of incorporating buserelin into semen and using it for insemination needs thorough investigation before large-scale implementation.

In our study, even under the challenging conditions involving gilts in a tropical climate, we confirmed the results demonstrated by Manjarin et al. [25]. Our data indicated that the treatment group receiving 10 µg (2.5 mL) of buserelin acetate significantly increased the litter size in gilts, both in total and live-born piglets, by an average of 0.8 to 0.9 piglets per litter. However, the dose of buserelin used in our study differs from the previous study. In the present study, we followed the manufacturer’s recommended dose of 10 µg for intramuscular injection to effectively induce ovulation in gilts and sows [2,13,17]. Additionally, we evaluated a half dose of 5 µg per gilt. The doses of buserelin used in our study are higher than those in the previous study by Manjarin et al. [25], which used 2 µg per gilt. We aimed to determine whether different doses, particularly higher ones, could effectively improve litter size in pigs. In the previous study, only one dose of buserelin was evaluated, and it was much lower than the recommended dose for inducing ovulation in sows [13,17]. These findings suggest that the optimal doses of buserelin for enhancing litter size in sows via artificial insemination, as well as associated factors, should be investigated further.

In the previous study [25], experiments were conducted on a farm located in a continental Mediterranean climate zone. Sow performance was monitored over three periods: January to April, May to August, and September to December, across two consecutive years. The accommodations for the sows included cooling panels to limit the maximum temperature to 28 °C. This study aimed to assess the effect of adding various hormones, including oxytocin, PGF2α, and buserelin, to boar semen doses on sow reproductive performance throughout the year. Results showed an improvement in the total number of piglets born per litter in sows inseminated with semen supplemented with buserelin in all seasons. Additionally, the farrowing rate improved from 84.9% in the control group to 93.2% in sows inseminated with buserelin-supplemented semen. In the present study, experiments were conducted under tropical conditions over a short period (four months) and included only gilts. All gilts were housed in evaporative cooling systems to reduce the effects of high outdoor temperatures. This system has been used in Thai swine herds for over 20 years and has now become the most common type of pig housing in Thailand [26]. During the experimental period, the average temperature inside the barn was 22.9 ± 3.6 °C, with a range of 14.5 to 32.0 °C, and average humidity levels were 88.8 ± 7.0%, ranging from 45.5 to 98.0%. In practice, the barn temperature was kept below 27 °C, though humidity was not controlled. However, the cooling systems used in pig farms have limited capacity, reducing the outdoor temperature by only 5–8 °C [26]. In Thailand, during the hot months of March, April, and May, outdoor temperatures can reach up to 40 °C, which can raise the temperature inside the barn to as high as 35 °C. This, combined with high humidity, can cause heat stress in pregnant gilts and compromise their reproductive performance [26]. The average farrowing rate observed in this study was relatively low at 78.8%, compared to a previous study in Spain, which reported rates of 84.9–93.2% [25]. This lower rate may be due to the inclusion of only gilts and the generally lower farrowing rates of gilts in tropical climates [27]. A prior study in Thailand found average farrowing rates of 73.1% for gilts, 81.7% for primiparous sows, and 84.9% to 85.9% for multiparous sows [27]. In that study [27], data were collected from 2005 to 2008, and sows were kept in open housing systems equipped with fans and water sprinklers to lower barn temperatures. The study reported average daily minimum–maximum outdoor temperatures of 21.1–33.3 °C, 24.4–31.6 °C, and 17.9–29.9 °C during the hot, rainy, and cool seasons, respectively. The 24 h average humidity levels were 68.3%, 81.7%, and 64.2% for the hot, rainy, and cool seasons, respectively. Despite the use of evaporative cooling systems in most pig farms, the poor farrowing rates observed in gilts under tropical climates for over 20 years [27] indicate that additional solutions are needed to improve reproductive performance under these conditions. In this study, the farrowing rate of gilts varied from 73.2% to 85.1%. Although the treatment groups showed a lower farrowing rate compared to the control group, the difference was not statistically significant, possibly due to the limited number of animals in each group. However, the observed reduction in farrowing rates in both treatment groups warrants further investigation with a larger sample size.

The increased litter size observed in the present study may be due to the induction of ovulation through the intravaginal delivery of buserelin [17]. This method ensures that ovulation aligns with the introduction of fresh semen into the reproductive system, creating optimal conditions for fertilization [5]. This likely leads to an increased fertilization rate of oocytes, resulting in a higher number of piglets born. Additionally, this method offers labor savings and enhances animal welfare. While the effects of triptorelin on ovulation have been studied [28,29], similar research on buserelin has not been previously conducted. Buserelin is known to trigger ovulation within 30–33 h after intramuscular administration [13], compared to the 44–48 h required by triptorelin. This difference highlights the potential for further research, especially considering the 24 h viability of sperm within the swine reproductive tract [30]. Given that the buserelin–semen mixture was applied only during the first artificial insemination, it is conceivable that fertilization primarily occurs with the subsequent insemination, 12 h later. Future research will explore the effects of intravaginal buserelin application followed by a single insemination 12–24 h later to further understand this phenomenon.

The present study found that using 10 µg of buserelin resulted in an increase of approximately 0.8–0.9 additional live-born piglets per litter. The economic value of these additional piglets can vary significantly depending on the country, herd, and routine investment. In Thailand, the cost of a piglet at weaning is about 44.4 USD. The cost of a GnRH agonist also varies by manufacturer and marketing policy, typically ranging from 2 to 10 USD per dose, which is substantially lower than the value of a piglet. However, it is important to consider the survival rate of piglets and other associated costs, which can vary widely among herds and are difficult to estimate. Despite these variables, comparing the cost of buserelin to the value of additional piglets suggests a net economic benefit. This highlights the potential for increasing economic returns by enhancing the efficacy of artificial insemination through precise ovulation control.

A GnRH agonist is a synthetic peptide that mimics the natural GnRH produced by the hypothalamus in the brain and is commonly used in both humans and animals [13,14,15,16,17,18,19]. GnRH plays a crucial role in regulating reproductive hormones in both males and females. In general, GnRH agonists have a relatively short half-life. Previous studies have shown that the half-life of buserelin in gilts averages 1.3 h, with a drug clearance rate of approximately 41.2 L per hour [31]. This indicates that most of the exogenous hormone is eliminated from the animal’s body within a short period. Buserelin, in particular, has been widely used in the bovine and swine industries for over three decades [13,17]. To our knowledge, there is no significant evidence of negative consequences associated with GnRH agonists in pigs or cattle. However, based on previous pharmacological studies, potential side effects on the next farrowing could not be anticipated [31].

Regarding semen evaluation after adding 5 or 10 µg of buserelin to the semen dose, it was observed that while total sperm motility did not change significantly, a decrease in progressive sperm motility was detected when 10 µg of buserelin was added. A previous study in rabbits demonstrated that another GnRH analog, lecirelin, diluted in various excipients (such as benzylic alcohol, benzoic acid, and paraben) and added to a seminal dose, led to different fertility rates and affected the percentage of capacitated sperm when inseminated intravaginally [32]. The lowest percentage of capacitated sperm and the lowest fertility rate were observed in rabbits inseminated with lecirelin diluted in benzylic alcohol [32]. In the current study, benzyl alcohol was used as an excipient in buserelin acetate. Benzyl alcohol is a compound frequently employed as a bacteriostatic preservative in various medications, cosmetics, and personal care products. Regarding its effects on sperm, benzyl alcohol has been shown to exhibit spermicidal properties, potentially damaging sperm cells and reducing their motility and viability [33]. This effect is primarily due to benzyl alcohol disrupting sperm cell membranes and interfering with their normal functions [33]. Consequently, exposure to benzyl alcohol, particularly at higher concentrations or with prolonged use, may negatively impact sperm quality and fertility. It is important to recognize that the degree of impact can vary depending on factors such as the concentration of benzyl alcohol, duration of exposure, and individual sensitivity. This suggests that the observed negative effect of 10 µg of buserelin acetate (2.5 mL) on sperm may be more closely related to the excipients used rather than the direct action of buserelin itself. Therefore, it is crucial to consider alternative excipients when diluting buserelin before insemination. This might also explain the numerical, though not statistically significant, reduction in both conception rate and farrowing rate observed after the addition of buserelin to the semen dose. However, further large-scale studies are necessary to fully understand the overall impact of buserelin acetate on reproductive efficiency. Additionally, prolonged direct exposure of buserelin acetate in benzyl alcohol to semen should be avoided.

Further areas for exploration were identified as well. It was observed that the treatment 1 group did not significantly differ from the control group in terms of the total number of piglets born per litter. Similarly, there were no significant differences in farrowing rates between the groups and the control. This indicates that the effectiveness of buserelin in inducing ovulation when combined with semen may be dose-dependent. It is worth noting that the manufacturer’s recommended dose for reliable ovulation induction via intramuscular injection is 10 µg of buserelin acetate [13]. This suggests the possibility that higher doses might yield improved outcomes. In rabbit studies, the effective dose of buserelin needed in semen for reliable ovulation induction was found to be 15% higher than the dose administered intramuscularly [34]. Therefore, future research could explore the impact of increased buserelin doses.

To the best of our understanding, this study pioneers an extension of previous research by examining the effects of adding buserelin to boar semen used for gilt insemination under the challenging conditions of a tropical climate. In conclusion, incorporating 10 µg of buserelin acetate into boar semen appears to increase litter size in gilts, resulting in an average increase of approximately 0.8 total-born piglets and 0.9 more live-born piglets per litter. Additional studies are needed to clarify the underlying processes and to refine the dosage and timing of buserelin supplementation.

## 5. Conclusions

The addition of buserelin to boar semen showed potential to improve reproductive outcomes in gilts. While farrowing rates remained similar across all groups, the use of 10 µg buserelin resulted in increased total piglets born per litter, more live-born piglets, and higher litter birth weights compared to the control and lower dose groups. These findings suggest that supplementing boar semen with 10 µg buserelin could enhance gilt litter size and birth weight, indicating a promising strategy for improving swine reproductive performance.

## Figures and Tables

**Table 1 animals-14-02501-t001:** Reproductive performances of gilts after buserelin supplementation in boar semen of 5 µg (treatment 1) or 10 µg (treatment 2) compared with conventional artificial insemination (control) (least-square means ± SEM).

Variables	Control	Treatment 1	Treatment 2	*p*-Value
Age of gilts (days)	221.6 ± 1.9	226.9 ± 1.9	224.9 ± 2.0	0.154
Backfat thickness (mm)	15.9 ± 0.3	15.1 ± 0.4	15.0 ± 0.4	0.161
Semen evaluation				
Total sperm motility (%)	84.0 ± 1.3	82.3 ± 1.3	80.5 ± 1.3	0.191
Progressive motility (%)	49.5 ± 3.2 ^a^	42.6 ± 3.2 ^ab^	37.7 ± 3.2 ^b^	0.042
Fertility traits				
Number of inseminated gilts	94	71	66	
Regular return-to-estrus (%)	4.3	5.6	10.6	0.259
Conception rate (%)	90.4	80.3	77.3	0.058
Farrowing rate (%)	85.1	73.2	75.8	0.141
Litter traits				
Number of litters	80	51	50	
Gestation length (days)	115.6 ± 0.1	115.6 ± 0.2	115.6 ± 0.2	0.919
Total number of piglets born per litter	13.2 ± 0.3 ^a^	13.3 ± 0.4 ^ab^	14.0 ± 0.3 ^b^	0.134
Number of piglets born alive per litter	12.3 ± 0.3 ^a^	12.0 ± 0.4 ^a^	13.2 ± 0.3 ^b^	0.035
Stillborn piglets (%)	3.9 ± 0.7 ^a^	5.7 ± 0.8 ^ab^	2.7 ± 0.9 ^b^	0.052
Mummified fetuses (%)	2.1 ± 1.0	5.0 ± 1.3	2.9 ± 1.3	0.207
Litter birth weight (kg)	15.6 ± 0.3 ^a^	15.7 ± 0.4 ^a^	17.0 ± 0.4 ^b^	0.019
Piglet characteristics				
Number of piglets	463	506	449	
Piglet birth weight (g)	1218 ± 24.9	1242 ± 24.1	1266 ± 25.5	0.409
Piglets with birth less than 1000 g (%)	15.0	15.2	11.1	0.146
Coefficient of variation of piglet birth Weight within the litter (%)	16.6 ± 1.3	16.6 ± 1.3	16.0 ± 1.3	0.934

*p*-value (upper right) indicates the overall significant effect of the main factors using analysis of variance (F statistic). The different letters (a, b) within a row indicate significant differences (*p* < 0.05) based on pairwise comparisons between group least-square means using the least significant difference (LSD) post hoc test.

## Data Availability

The data presented in this study are available on request from the corresponding author.

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
