# Peer review of "Buserelin Acetate Added to Boar Semen Enhances Litter Size in Gilts in Tropical Environments"

_animals, 2024, doi:10.3390/ani14172501_

Round 1
Reviewer 1 Report
Comments and Suggestions for Authors
The manuscript entitled "Buserelin acetate added to boar semen enhances litter size in gilts” is well written and covers an important theme dedicated to the efficacy of swine production. This is a methodological work: gilts were artificially inseminated by sperm which contained different concentrations of buserelin. In the work a large number of animals were used, therefore there is no doubt about the reliability of the data obtained.
In general, it is a well done work, however I have some questions.
General questions:
1. For why it is necessary to dilute the buserelin in semen? It seems that it will be easier to make an injection for swine? Discuss it.
2. The main idea is that is better that application of buserelin is good because it leads to the increase on the numbers of piglets at farrowing. Are you sure that it is true? You provide data about the weight of piglets, but what about other indicators of their health? How many of these “buserelin-concieved” piglets died at one month and half of the year? How rapidly these piglets gained weight in comparison to the control group? Etc?
3. Discuss in details economical effect. Which is the cost of buserelin? Which is the cost of additional 0.8-0.9 piglets obtained after buserelin treatment? Is the application of buserelin economically reasonable in comparison to standard natural conception or artificial insemination?
4. What was the state of that sows which were treated by buserelin? Did this treatment affect to the number of piglets in their next farrowings? If you do not have these data you should discuss the potential side effects of buserelin application.
5. You mention that your work is the replication of the work of Manjarin et al [24]. However, in the work of Manjarin buserelin at concentration 2 µg per dose was used. You use another concentrations. Discuss in details observed differences.
Minor questions:
1. Why did you used sows only at 2nd o 3rd estrus?
2. Storage of semen at +17oC is a typical prtocol? line 169 If yes, provide some references. If not, discuss why you used this protocol.
3. Provide some photos of piglets from control and “buserelin-concieved” groups.
4. Provide video with motile sperm which was used for insemination.
Author Response
# Reviewer 1
Comments and Suggestions for Authors
The manuscript entitled "Buserelin acetate added to boar semen enhances litter size in gilts” is well written and covers an important theme dedicated to the efficacy of swine production. This is a methodological work: gilts were artificially inseminated by sperm which contained different concentrations of buserelin. In the work a large number of animals were used, therefore there is no doubt about the reliability of the data obtained.
In general, it is a well done work, however I have some questions.
Response: Thank you very much for your kind consideration and efforts to improve the quality of our manuscript. We have carefully considered all the comments and suggestions, and the manuscript has been revised accordingly. All changes in the manuscript are indicated in blue text.
General questions:
- For why it is necessary to dilute the buserelin in semen? It seems that it will be easier to make an injection for swine? Discuss it.
Response: Additional discussion has been added: “This study aims to dilute buserelin in semen and administer it into the uterus simultaneously with artificial insemination, as an alternative to the traditional intramuscular injection route. This method was explored to reduce the need for injections and improve animal welfare in the swine industry. Intramuscular injections often cause stress and discomfort to animals, which can negatively affect their reproductive outcomes. Such injections should be limited to sick animals and not used routinely, such as for artificial insemination. Previous research has shown that another GnRH agonist, Triptorelin acetate, can be administered intravaginally to induce ovulation in sows [22]. However, the feasibility of incorporating buserelin into semen and using it for insemination needs thorough investigation before large-scale implementation.”
- The main idea is that is better that application of buserelin is good because it leads to the increase on the numbers of piglets at farrowing. Are you sure that it is true? You provide data about the weight of piglets, but what about other indicators of their health? How many of these “buserelin-concieved” piglets died at one month and half of the year? How rapidly these piglets gained weight in comparison to the control group? Etc?
Response: “The primary objective of this study is to investigate the effectiveness of buserelin in inducing ovulation when combined with semen. Litter size at birth is the main indicator used for this purpose. Additionally, we assessed the number of live-born piglets and their birth weights. However, a limitation of our study is that we could not track piglet mortality rates during the lactation, nursery, and fattening periods due to management issues such as cross-fostering and mixing animals during the nursery phases. Therefore, the efficacy of buserelin in inducing ovulation in gilts can only be demonstrated by an increase in litter size at birth, which is also an important economic trait in pigs.” This issue has been added into the discussion part. Thank you very much.
- Discuss in details economical effect. Which is the cost of buserelin? Which is the cost of additional 0.8-0.9 piglets obtained after buserelin treatment? Is the application of buserelin economically reasonable in comparison to standard natural conception or artificial insemination?
Response: “The study found that using 10 µg of buserelin resulted in an increase of approximately 0.8-0.9 additional live-born piglets per litter. The economic value of these additional piglets can vary significantly depending on the country, herd, and routine investment. In Thailand, the cost of a piglet at weaning is about 44.4 USD. The cost of a GnRH agonist also varies by manufacturer and marketing policy, typically ranging from 2 to 10 USD per dose, which is substantially lower than the value of a piglet. However, it is important to consider the survival rate of piglets and other associated costs, which can vary widely among herds and are difficult to estimate. Despite these variables, comparing the cost of buserelin to the value of additional piglets suggests a net economic benefit. This highlights the potential for increasing economic returns by enhancing the efficacy of artificial insemination through precise ovulation control.” Additional discussion concerning this issue has been added to the manuscript.
- What was the state of that sows which were treated by buserelin? Did this treatment affect to the number of piglets in their next farrowings? If you do not have these data you should discuss the potential side effects of buserelin application.
Response: “A GnRH agonist is a synthetic peptide that mimics the natural GnRH produced by the hypothalamus in the brain and is commonly used in both humans and animals [13 – 19]. GnRH plays a crucial role in regulating reproductive hormones in both males and females. In general, GnRH agonists have a relatively short half-life. Previous studies have shown that the half-life of buserelin in gilts averages 1.3 hours, with a drug clearance rate of approximately 41.2 liters per hour [32]. This indicates that most of the exogenous hormone is eliminated from the animal's body within a short period. Buserelin, in particular, has been widely used in the bovine and swine industries for over three decades [13,17]. To our knowledge, there is no significant evidence of negative consequences associated with GnRH agonists in pigs or cattle. However, based on previous pharmacological studies, potential side effects on the next farrowing could not be anticipated [32].” Additional discussion on this issue has been added.
- You mention that your work is the replication of the work of Manjarin et al [24]. However, in the work of Manjarin buserelin at concentration 2 µg per dose was used. You use another concentrations. Discuss in details observed differences.
Response: “In this study, we followed the manufacturer's recommended dose of 10 µg for intramuscular injection to effectively induce ovulation in gilts and sows [2,13,17]. Additionally, we evaluated a half dose of 5 µg per gilt. The doses of buserelin used in our study are higher than those in a previous study by Manjarin et al. [25] which used 2 µg per gilt. We aimed to determine whether different doses, particularly higher ones, could effectively improve litter size in pigs. In the previous study, only one dose of buserelin was evaluated, and it was much lower than the recommended dose for inducing ovulation in sows [13,17]. These findings suggest that the optimal doses of buserelin to enhance litter size in sows via artificial insemination, as well as associated factors, should be investigated further.” Additional discussion has been added.
Minor questions:
- Why did you used sows only at 2nd or 3rd estrus?
Response: The statement has been revised to “Mating was conducted once the gilts exhibited estrus symptoms following the conclusion of altrenogest treatment, provided they weighed at least 130 kg.” Thank you.
- Storage of semen at +17 °C is a typical protocol? line 169 If yes, provide some references. If not, discuss why you used this protocol.
Response: Generally, diluted boar semen is recommended to be kept at 15–20°C for up to 3 days before use. In our case, all diluted semen was utilized within 24 hours of collection to avoid variations in semen quality that can occur when preserved for 2–3 days. The statement has been revised to “The diluted semen was stored in a refrigerator at 17°C and used within one day of collection, in accordance with herd regulations [15].” An additional reference has also been added.
- Provide some photos of piglets from control and “buserelin-concieved” groups.
Response: Below is a photo of a sow and her piglets following insemination with boar semen supplemented with buserelin.
Figure 1. A sow and her piglets following insemination with boar semen supplemented with buserelin.
- Provide video with motile sperm which was used for insemination.
Response: We did not record a video of the motile sperm in the semen dose after adding buserelin, as the buserelin was added during the insemination process. However, we evaluated sperm motility in our laboratory after adding buserelin. The average total sperm motility, as determined by CASA, was 82.3 ± 1.3% with 5 µg of buserelin and 80.5 ± 1.3% with 10 µg of buserelin.

Reviewer 2 Report
Comments and Suggestions for Authors
The manuscript entitled: Buserelin acetate added to boar semen enhances litter size in gilts, was submitted for review.
The authors approach a theme of efficiency of fertility by controlling reproduction and fertility, by adding the first dose of AI with buserelin acetate.
It is known that using a gonadotropin-releasing hormone (GnRH) agonist, such as buserelin, at the onset of estrus can induce ovulation and reduce variation in ovulation time among sows.
This study investigated the effect of adding buserelin to boar semen on piglet size at farrowing. Females were inseminated with semen spiked with either 5 or 10 µg of buserelin acetate during initial insemination. Although the farrowing rate showed no significant differences between groups, the total number of piglets and piglets born alive was higher in the group receiving 10 µg of buserelin acetate. In addition, this group presented a higher fertility weight at birth compared to the control group. The authors state that the findings indicate that supplementation of boar sperm with a GnRH agonist increases litter size in gilts in a tropical environment.
General aspects:
The study seems well designed, from the summary, introduction, material and method, results and discussions. But technical and drafting corrections are necessary, attention to the writing of sperm concentrations.
Required changes:
The title does not exactly reflect the activities in the subject of the manuscript. The objective of the study is focused on birth for gilts in a tropical environment. I recommend changing / adapting the title.
The essential recommendation when presenting the subject in the introduction chapter, but also in the discussions, is to physiologically describe the pharmacodynamic effect that occurs through the intrauterine administration of the GnRH homologue.
Regarding the material and method, clarifications are needed on the AI method, was it intrauterine or intracervical (classic). IA Intrauterine uses several types of catheters. I recommend studying several articles in this regard: DOI: 10.1016/s0093-691x(02)00648-9
https://www.business.qld.gov.au/industries/farms-fishing-forestry/agriculture/animal/industries/pigs/breed/inseminate/ai-techniques.
There is not much information about the quality of the sperm from the boars used. I would like the details of the spermogram from the age of the boars, the volume of the ejaculate, types of mobility, concentration / ejaculated and per ml, and biosecurity protocols for the collection of boar sperm. I recommend the publication: DOI: 10.3390/molecules26206183
In Table 1, no data are entered about the quality of sperm at AI, (volume, concentration...)
The necessary information is available at:
- the mechanism of intrauterine GnRH administration, from the possible endometrial absorption and the follicular/ovarian effect,
- Clarifications of cervical or uterine AI biotechnique , in bought groups (control gr AI by clasic method?? L 225). info is missing from the material and method!
- Improvements in the quality of the sperm used to prepare AI doses, and the biosecurity of its collection and processing.
Author Response
# Reviewer 2
The manuscript entitled: Buserelin acetate added to boar semen enhances litter size in gilts, was submitted for review. The authors approach a theme of efficiency of fertility by controlling reproduction and fertility, by adding the first dose of AI with buserelin acetate. It is known that using a gonadotropin-releasing hormone (GnRH) agonist, such as buserelin, at the onset of estrus can induce ovulation and reduce variation in ovulation time among sows.
This study investigated the effect of adding buserelin to boar semen on piglet size at farrowing. Females were inseminated with semen spiked with either 5 or 10 µg of buserelin acetate during initial insemination. Although the farrowing rate showed no significant differences between groups, the total number of piglets and piglets born alive was higher in the group receiving 10 µg of buserelin acetate. In addition, this group presented a higher fertility weight at birth compared to the control group. The authors state that the findings indicate that supplementation of boar sperm with a GnRH agonist increases litter size in gilts in a tropical environment.
General aspects:
The study seems well designed, from the summary, introduction, material and method, results and discussions. But technical and drafting corrections are necessary, attention to the writing of sperm concentrations.
Required changes:
The title does not exactly reflect the activities in the subject of the manuscript. The objective of the study is focused on birth for gilts in a tropical environment. I recommend changing / adapting the title.
Response: The title has been modified to: “Buserelin acetate added to boar semen enhances litter size in gilts in tropical environments”
The essential recommendation when presenting the subject in the introduction chapter, but also in the discussions, is to physiologically describe the pharmacodynamic effect that occurs through the intrauterine administration of the GnRH homologue.
Response: Additional discussion concerning the route of administration has been added: “The secondary objective of the present study is to demonstrate the feasibility of diluting buserelin in semen and administering it into the uterus simultaneously with artificial insemination as an alternative to the traditional intramuscular injection route. This method was explored to reduce the need for injections and improve animal welfare in the swine industry. Intramuscular injections often cause stress and discomfort to animals, which can negatively affect their reproductive outcomes. Such injections should be limited to sick animals and not used routinely, such as for artificial insemination. Previous research has shown that another GnRH agonist, Triptorelin acetate, can be administered intravaginally to induce ovulation in sows [12,22]. However, the feasibility of incorporating buserelin into semen and using it for insemination needs thorough investigation before large-scale implementation.”
Regarding the material and method, clarifications are needed on the AI method, was it intrauterine or intracervical (classic). IA Intrauterine uses several types of catheters. I recommend studying several articles in this regard: DOI: 10.1016/s0093-691x(02)00648-9
Response: In the present study, intrauterine AI catheter was used. The statement on this issue has been mentioned in Materials and methods: “For artificial insemination (AI), an intrauterine AI catheter was used to insert a semen dose of 3.0 × 109 motile sperm in an 80 ml volume. The semen used for all the gilts was pooled from two to three Duroc boars, aged 1 to 3 years, and collected using the gloved hand technique. This semen was collected and processed daily at a boar stud, with each boar providing a semen sample every 5 to 7 days. For AI, pooled semen from two to three boars was used for each gilt.”
https://www.business.qld.gov.au/industries/farms-fishing-forestry/agriculture/animal/industries/pigs/breed/inseminate/ai-techniques.
There is not much information about the quality of the sperm from the boars used. I would like the details of the spermogram from the age of the boars, the volume of the ejaculate, types of mobility, concentration / ejaculated and per ml, and biosecurity protocols for the collection of boar sperm. I recommend the publication: DOI: 10.3390/molecules26206183
Response: Addition data concerning the boar semen quality used has been added in Materials and methods and “Results”:
“The semen used for all the gilts was pooled from two to three Duroc boars, aged 1 to 3 years, and collected using the gloved hand technique.”
“To assess the effect of buserelin on boar semen quality, an additional trial was conducted using 10 semen ejaculates from 10 Duroc boars. Each ejaculate was diluted with Beltsville Thawing Solution (BTS) and divided into three parts. These parts were then supplemented with either 0 (control), 5, or 10 µg of buserelin. Total sperm motility and progressive motility were measured within 20 min of adding buserelin, utilizing a computer-assisted sperm analysis system (SCA® CASA System, MICROPTIC S.L., Bar-celona, Spain).”
“In the analysis of boar semen quality, total sperm motility within 20 min after treatment with 0, 5, and 10 µg of buserelin did not differ significantly among the groups. The total sperm motility rates were 84.0 ± 1.3%, 82.3 ± 1.3%, and 80.5 ± 1.3%, respectively (P = 0.191). However, the progressive sperm motility of semen supplemented with 10 µg of buserelin significantly declined compared to the control group (37.7 ± 3.2% vs. 49.5 ± 3.2%, respectively, P = 0.033). Meanwhile, the progressive sperm motility of semen sup-plemented with 5 µg of buserelin did not differ significantly from the control group (42.6 ± 3.2%, P = 0.279).”
In Table 1, no data are entered about the quality of sperm at AI, (volume, concentration...)
Response: Total sperm motility and progressive motility has been added into Table 1.
The necessary information is available at:
- the mechanism of intrauterine GnRH administration, from the possible endometrial absorption and the follicular/ovarian effect,
Response: The exact mechanism of intrauterine GnRH administration still requires further study. However, we added some information concerning the use of this technique in rabbit “In rabbits, a species that requires induction for ovulation, the inclusion of GnRH agonists like buserelin and lecirelin in semen has been shown to enhance ovulation rates compared to controls [23].” Also additional discussion concerning this issue has been addressed.
- Clarifications of cervical or uterine AI biotechnique, in bought groups (control gr AI by classic method?? L 225). info is missing from the material and method!
Response: “Traditional AI” has been changed to “AI without buserelin supplementation”.
- Improvements in the quality of the sperm used to prepare AI doses, and the biosecurity of its collection and processing.
Response: Some information regarding semen quality has been added.
Reviewer 3 Report
Comments and Suggestions for Authors
The authors investigated the effects of buserelin supplementation to boar semen for artificial insemination on fertility trait, litter trait and piglet characteristics, and concluded that the supplementation of buserelin to boar semen increased total piglets born per litter, more live-born piglets, and higher litter birth weights.
General comments:
As the authors explained (Line 100-102, ref. 24), similar examination was done in the previous study, and the main focus of the present study can be to perform it in the tropical environment (Line 288-290). Nevertheless, the authors did not explain and discuss the difference of climate condition and results between previous (ref. 24) and present situation. Thus, the significance of the present results is unclear.
Specific comments:
Line 59: Please explain whether the authors think “multiple inseminations” should be avoided for efficiency or not.
Line 129-130: Please state the (setting) temperature and humidity of the closed breeding system.
Line 165: Please provide the actual sperm motility used.
Line 168: Was 37.5 × 106 sperm per ml motile or total sperm?
Line 247-257: This part is a duplicate of the Introduction section and should be deleted.
Table 1: What does “P value (upper right)” indicate? How is it different form “a,b Different letters within a row indicate significant differences (P < 0.05)”
Author Response
# Reviewer 3
Comments and Suggestions for Authors
The authors investigated the effects of buserelin supplementation to boar semen for artificial insemination on fertility trait, litter trait and piglet characteristics, and concluded that the supplementation of buserelin to boar semen increased total piglets born per litter, more live-born piglets, and higher litter birth weights.
General comments:
As the authors explained (Line 100-102, ref. 24), similar examination was done in the previous study, and the main focus of the present study can be to perform it in the tropical environment (Line 288-290). Nevertheless, the authors did not explain and discuss the difference of climate condition and results between previous (ref. 24) and present situation. Thus, the significance of the present results is unclear.
Response: In the previous study [25], experiments were conducted on a farm located in a continental-Mediterranean climate zone. Sow performance was monitored over three periods: January to April, May to August, and September to December, across two consecutive years. The accommodations for the sows included cooling panels to limit the maximum temperature to 28°C. This study aimed to assess the effect of adding various hormones, including oxytocin, PGF2α, and buserelin, to boar semen doses on sow reproductive performance throughout the year. Results showed an improvement in the total number of piglets born per litter in sows inseminated with semen supplemented with buserelin in all seasons. Additionally, the farrowing rate improved from 84.9% in the control group to 93.2% in sows inseminated with buserelin-supplemented semen. In the present study, experiments were conducted under tropical conditions over a short period (four months) and included only gilts. All gilts were housed in evaporative cooling systems to mitigate the impact of high outdoor temperatures. The average temperature inside the barn was 22.9 ± 3.6°C, ranging from 14.5 to 32.0°C, and the average humidity levels were 88.8 ± 7.0%, with a range of 45.5 to 98.0%. The average farrowing rate observed in the present study was relatively low (78.8%) compared to the previous study in Spain (84.9 – 93.2%) [25], due to the inclusion of only gilts and the generally lower farrowing rate of gilts under tropical climates [27]. A previous study in Thailand demonstrated average farrowing rates of 73.1% for gilts, 81.7% for primiparous sows, and 84.9% to 85.9% for multiparous sows [27]. Due to the poor farrowing rates observed in gilts under tropical climates [27], the present study focused on improving their reproductive performance. The farrowing rate of gilts in the present study varied from 73.2% to 85.1%. Although the treatment groups showed a lower farrowing rate compared to the control group, the difference was not statistically significant, possibly due to the limited number of animals in each group. Nonetheless, the reduction in farrowing rate observed in both treatment groups warrants further investigation with a larger sample size.
Specific comments:
Line 59: Please explain whether the authors think “multiple inseminations” should be avoided for efficiency or not.
Response: Additional information has been added: “Indeed, the ovulation process in sows occurs within a short period of time (i.e., within 3 h) [4], making it crucial to inseminate with a single dose of semen close to the time of ovulation [5]. Consequently, multiple inseminations (2-3 times during standing estrus) are not entirely necessary if the timing of ovulation can be controlled.”.
Line 129-130: Please state the (setting) temperature and humidity of the closed breeding system.
Response: We have added,” The temperatures inside the barn averaged 22.9 ± 3.6°C, ranging from 14.5 to 32.0°C. The average humidity levels were 88.8 ± 7.0%, with a range of 45.5 to 98.0%.” in Materials and methods as suggested.
Line 165: Please provide the actual sperm motility used.
Response: Actual sperm motility has been added. “The total sperm motility of boar semen in the present study averaged 82.3 ± 4.3% (mean ± SD).”
Line 168: Was 37.5 × 106 sperm per ml motile or total sperm?
Response: The statement has been revised to “37.5 × 106 motile sperm per ml”.
Line 247-257: This part is a duplicate of the Introduction section and should be deleted.
Response: Deleted as suggested.
Table 1: What does “P value (upper right)” indicate? How is it different form “a,b Different letters within a row indicate significant differences (P < 0.05)”.
Response: ‘P value (upper right) indicates the overall significant effect of the main factors using analysis of variance (F statistic). The different letters (a, b) within a row indicated significant differences based on a pairwise comparisons between group least-square means by using least significant difference (LSD) post hoc test.’ This statement has been included in the footnote of the Table.
Round 2
Reviewer 3 Report
Comments and Suggestions for Authors
General comments:
As the authors explained (Line 100-102, ref. 24), similar examination was done in the previous study, and the main focus of the present study can be to perform it in the tropical environment (Line 288-290). Nevertheless, the authors did not explain and discuss the difference of climate condition and results between previous (ref. 24) and present situation. Thus, the significance of the present results is unclear.
Response: --- In the present study, experiments were conducted under tropical conditions over a short period (four months) and included only gilts. All gilts were housed in evaporative cooling systems to mitigate the impact of high outdoor temperatures. The average temperature inside the barn was 22.9 ± 3.6°C, ranging from 14.5 to 32.0°C, and the average humidity levels were 88.8 ± 7.0%, with a range of 45.5 to 98.0%. ---due to the inclusion of only gilts and the generally lower farrowing rate of gilts under tropical climates [27].
>Is it common to keep pigs in barn with evaporative cooling system in Thai farms?
I like to confirm that the present housing condition represents a tropical climate situation, since it can reach to 32 °C in summer even in Spain.
A previous study in Thailand demonstrated average farrowing rates of 73.1% for gilts, 81.7% for primiparous sows, and 84.9% to 85.9% for multiparous sows [27]. Due to the poor farrowing rates observed in gilts under tropical climates [27], the present study focused on improving their reproductive performance.
>Was the housing condition in previous report [27] the same to that in the present study (housed in evaporative cooling systems)?
Specific comments:
Line 129-130: Please state the (setting) temperature and humidity of the closed breeding system.
Response: We have added,” The temperatures inside the barn averaged 22.9 ± 3.6°C, ranging from 14.5 to 32.0°C. The average humidity levels were 88.8 ± 7.0%, with a range of 45.5 to 98.0%.” in Materials and methods as suggested.
Line 165: Please provide the actual sperm motility used.
Response: Actual sperm motility has been added. “The total sperm motility of boar semen in the present study averaged 82.3 ± 4.3% (mean ± SD).”
>Please discuss why total and progressive sperm motility, conception and farrowing rate decreased by adding buserelin, and their impact on overall reproductive efficiency.
Table 1: What does “P value (upper right)” indicate? How is it different form “a,b Different letters within a row indicate significant differences (P < 0.05)”.
Response: ‘P value (upper right) indicates the overall significant effect of the main factors using analysis of variance (F statistic). The different letters (a, b) within a row indicated significant differences based on a pairwise comparisons between group least-square means by using least significant difference (LSD) post hoc test.’ This statement has been included in the footnote of the Table.
>I am not sure whether it is a proper statistical procedure to perform a post hoc test (LSD) to show a significant difference when there is no significant difference by the F statistic (Total number of piglets born per litter, Stillborn piglets).
Author Response
# Reviewer 1
General comments:
As the authors explained (Line 100-102, ref. 24), similar examination was done in the previous study, and the main focus of the present study can be to perform it in the tropical environment (Line 288-290). Nevertheless, the authors did not explain and discuss the difference of climate condition and results between previous (ref. 24) and present situation. Thus, the significance of the present results is unclear.
Response: --- In the present study, experiments were conducted under tropical conditions over a short period (four months) and included only gilts. All gilts were housed in evaporative cooling systems to mitigate the impact of high outdoor temperatures. The average temperature inside the barn was 22.9 ± 3.6°C, ranging from 14.5 to 32.0°C, and the average humidity levels were 88.8 ± 7.0%, with a range of 45.5 to 98.0%. ---due to the inclusion of only gilts and the generally lower farrowing rate of gilts under tropical climates [27].
>Is it common to keep pigs in barn with evaporative cooling system in Thai farms?
Response: Yes, it is common to house pigs in barns with evaporative cooling systems in Thailand. This technology has been part of the Thai swine industry for over 20 years. Currently, majority of the Thai swine herds utilize evaporative cooling systems. Additional discussion has been added: “In the present study, experiments were conducted under tropical conditions over a short period (four months) and included only gilts. All gilts were housed in evaporative cooling systems to reduce the effects of high outdoor temperatures. This system has been used in Thai swine herds for over 20 years and has now become the most common type of pig housing in Thailand. During the experimental period, the average temperature inside the barn was 22.9 ± 3.6°C, with a range of 14.5 to 32.0°C, and average humidity levels were 88.8 ± 7.0%, ranging from 45.5 to 98.0%. In practice, the barn temperature was kept below 27°C, though humidity was not controlled. However, the cooling systems used in pig farms have limited capacity, reducing the outdoor temperature by only 5-8°C [26]. In Thailand, during the hot months of March, April, and May, outdoor temperatures can reach up to 40°C, which can raise the temperature inside the barn to as high as 35°C. This, combined with high humidity, can cause heat stress in pregnant gilts and compromise their reproductive performance [26]. The average farrowing rate observed in this study was relatively low at 78.8%, compared to a previous study in Spain, which reported rates of 84.9–93.2% [25]. This lower rate may be due to the inclusion of only gilts and the generally lower farrowing rates of gilts in tropical climates [27]. A prior study in Thailand found average farrowing rates of 73.1% for gilts, 81.7% for primiparous sows, and 84.9% to 85.9% for multiparous sows [27]. In that study [27], data were collected from 2005 to 2008, and sows were kept in open housing systems equipped with fans and water sprinklers to lower barn temperatures. The study reported average daily mini-mum-maximum outdoor temperatures of 21.1–33.3°C, 24.4–31.6°C, and 17.9–29.9°C during the hot, rainy, and cool seasons, respectively. The 24-hour average humidity levels were 68.3%, 81.7%, and 64.2% for the hot, rainy, and cool seasons, respectively. Despite the use of evaporative cooling systems in most pig farms, the poor farrowing rates observed in gilts under tropical climates for over 20 years [27] indicate that additional solutions are needed to improve reproductive performance under these conditions. In this study, the farrowing rate of gilts varied from 73.2% to 85.1%. Although the treatment groups showed a lower farrowing rate compared to the control group, the difference was not statistically significant, possibly due to the limited number of animals in each group. However, the observed reduction in farrowing rates in both treatment groups warrants further investigation with a larger sample size.”
I like to confirm that the present housing condition represents a tropical climate situation, since it can reach to 32 °C in summer even in Spain.
Response: Yes, the evaporative cooling systems used in pig farms have limited capacity, reducing the outdoor temperature by only 5-8°C. In Thailand, during the hot months of March, April, and May, outdoor temperatures can reach up to 40°C, raising the temperature inside the barn to as much as 35°C. However, during the experimental period, the temperatures were not excessively high, and the conditions inside the barn remained acceptable.
A previous study in Thailand demonstrated average farrowing rates of 73.1% for gilts, 81.7% for primiparous sows, and 84.9% to 85.9% for multiparous sows [27]. Due to the poor farrowing rates observed in gilts under tropical climates [27], the present study focused on improving their reproductive performance.
>Was the housing condition in previous report [27] the same to that in the present study (housed in evaporative cooling systems)?
Response: “In the previous study [27], data were collected from 2005 to 2008 and sows were kept in open housing systems equipped with fans and water sprinklers to reduce the temperature inside the barn. The study reported average daily minimum-maximum outdoor temperatures of 21.1–33.3°C, 24.4–31.6°C, and 17.9–29.9°C during the hot, rainy, and cool seasons, respectively. The 24-hour average humidity levels were 68.3%, 81.7%, and 64.2% for the hot, rainy, and cool seasons, respectively.“ This information has been added in the discussion section.
Specific comments:
Line 129-130: Please state the (setting) temperature and humidity of the closed breeding system.
Response: “In the closed housing system, the temperature inside the barn was maintained below 27°C, while humidity was not regulated.” This information has been added to the manuscript.
Response: We have added,” The temperatures inside the barn averaged 22.9 ± 3.6°C, ranging from 14.5 to 32.0°C. The average humidity levels were 88.8 ± 7.0%, with a range of 45.5 to 98.0%.” in Materials and methods as suggested.
Line 165: Please provide the actual sperm motility used.
Response: Actual sperm motility has been added. “The total sperm motility of boar semen in the present study averaged 82.3 ± 4.3% (mean ± SD).”
>Please discuss why total and progressive sperm motility, conception and farrowing rate decreased by adding buserelin, and their impact on overall reproductive efficiency.
Response: Additional discussion has been added. “Regarding semen evaluation after adding 5 or 10 µg of buserelin to the semen dose, it was observed that while total sperm motility did not change significantly, a decrease in progressive sperm motility was detected when 10 µg of buserelin was added. A previous study in rabbits demonstrated that another GnRH analog, Lecirelin, diluted in various excipients (such as benzilic alcohol, benzoic acid, and paraben) and added to a seminal dose, led to different fertility rates and affected the percentage of capacitated sperm when inseminated intravaginally [32]. The lowest percentage of capacitated sperm and the lowest fertility rate were observed in rabbits inseminated with Lecirelin diluted in benzilic alcohol [32]. In the current study, benzyl alcohol is used as an excipient in buserelin acetate. Benzyl alcohol is a compound frequently employed as a bacteriostatic preservative in various medications, cosmetics, and personal care products. Regarding its effects on sperm, benzyl alcohol has been shown to exhibit spermicidal properties, potentially damaging sperm cells and reducing their motility and viability [33]. This effect is primarily due to benzyl alcohol disrupting sperm cell membranes and interfering with their normal functions [33]. Consequently, exposure to benzyl alcohol, particularly at higher concentrations or with prolonged use, may negatively impact sperm quality and fertility. It's important to recognize that the degree of impact can vary depending on factors such as the concentration of benzyl alcohol, duration of exposure, and individual sensitivity. This suggests that the observed negative effect of 10 µg of buserelin acetate (2.5 mL) on sperm may be more closely related to the excipients used rather than the direct action of buserelin itself. Therefore, it is crucial to consider alternative excipients when diluting buserelin before insemination. This might also explain the numerical, though not statistically significant, reduction in both conception rate and farrowing rate observed after the addition of buserelin to the semen dose. However, further large-scale studies are necessary to fully understand the overall impact of buserelin acetate on reproductive efficiency. Additionally, prolonged direct exposure of buserelin acetate in benzyl alcohol to semen should be avoided.”
Table 1: What does “P value (upper right)” indicate? How is it different form “a,b Different letters within a row indicate significant differences (P < 0.05)”.
Response: ‘P value (upper right) indicates the overall significant effect of the main factors using analysis of variance (F statistic). The different letters (a, b) within a row indicated significant differences based on a pairwise comparisons between group least-square means by using least significant difference (LSD) post hoc test.’ This statement has been included in the footnote of the Table.
>I am not sure whether it is a proper statistical procedure to perform a post hoc test (LSD) to show a significant difference when there is no significant difference by the F statistic (Total number of piglets born per litter, Stillborn piglets).
Response: The post hoc test (LSD) was conducted to indicate a significant difference when the P value from the F statistic was below 0.05, or to show a tendency when P was less than or equal 0.1. We did not conduct the post hoc test at all when the P value was too far from significant.
